# Understanding antibiotic use in the community setting in Thailand: Does communication matter?

**Malee Sunpuwan**[1]*, **Sureeporn Punpuing**[1], **Wipaporn Jaruruengpaisan**[1]*, **Heiman Wertheim**[2]

**1** Institute for Population and Social Research, Mahidol University, Phutthamonthon, Thailand, **2** Department of Clinical Microbiology, Radboud Center for Infectious Diseases, Radboudumc, Nijmegen, Netherlands

* malee.sun@mahidol.ac.th (MS); wipaporn.jau@mahidol.ac.th (WJ)

**Data Availability Statement:** All relevant data are within the article and its Supporting information files. The coding data from the study conducted at the Kanchanaburi Health and Demographic

## Abstract

### Background

It is known that the misuse and overuse of antimicrobials leads to antimicrobial resistance (AMR). Effective communication between dispensers and users is thus crucial in reducing inappropriate antibiotic use.

### Objective

This study aims to gain a better understanding of communication around the use of antibiotics in the community and seeks potential implementation strategies to change dispenser and user practices in communication aspects.

### Methods

Qualitative methods were employed, including in-depth interviews with 18 drug suppliers and 16 community members, and eight focus group discussions with key informants. Data were collected in the Kanchanaburi Demographic Health Surveillance System in urban and semi-urban communities in the western region of Thailand. The thematic analysis included communication quality, communication and imbalanced power, and misconceptions and instruction. The OpenCode qualitative software program was employed.

### Results

The study revealed that the quality of communication was significantly influenced by the interaction of antibiotic dispensing with language and information. This interaction creates communication constraints between those dispensing antibiotics and the recipients, resulting in a less-than-optimal exchange of information. Consequently, users received limited information concerning the proper use of antibiotics. Furthermore, power imbalances and communication dynamics were perpetuated, mainly stemming from varying levels of access to and knowledge about antibiotics. This imbalance in power dynamics became evident between those dispensing antibiotics and the users. Users, as well as dispensers lacking

Surveillance System (HDSS) in Thailand is available on Harvard Dataverse (https://doi.org/10.7910/DVN/LMISZU).

**Funding:** 1.The Wellcome Trust, UK 2. The Institute for Population and Social Research, Mahidol University, Thailand.

**Competing interests:** The authors have declared that no competing interests exist.

proper qualifications, found themselves in a precarious position due to their inadequate knowledge of antibiotics. Moreover, it is noteworthy that misconceptions often conflicted with antibiotic instructions, leading to challenges in adhering to antibiotic regimens. These challenges primarily arose from misconceptions about antibiotics and concerns about potential side effects, particularly when users started to feel better.

## Conclusions

The findings highlight the importance of enhancing communication between dispensers and users through future interventions. These interventions should aim to bolster user understanding of antibiotics and provide clear, trustworthy instructions for their proper usage. Investigating innovative communication methods, such as the use of QR codes, presents a promising avenue for consideration. By addressing these communication gaps, we can advocate for the appropriate utilization of antibiotics and mitigate the prevalence of AMR.

## Introduction

The threat of antimicrobial resistance (AMR) to public health is of utmost concern, considering the diminishing effectiveness of antibiotics worldwide [1]. Thailand, like many other countries, is not exempt from this threat [2, 3]. In response, the Thai government has initiated various policies and programs aimed at improving antibiotic prescribing practices, increasing public awareness of appropriate antibiotic use and understanding AMR [3–5]. Notably, the "Antibiotics Smart Use (ASU) program" was launched in 2017, with the goal of reshaping antibiotic prescription practices and promoting responsible antibiotic use. Community education campaigns utilizing diverse educational resources such as infographic posters and pamphlets have been deployed to educate healthcare professionals and the general public [4, 5]. These concerted efforts have demonstrated promising results in curbing unnecessary antibiotic use [6] and elevating awareness levels [7]. Furthermore, national surveys tracking public knowledge and awareness of AMR and proper antibiotic use have indicated incremental improvements [8]. Nevertheless, antibiotic misuse is still challenging.

The challenge of antibiotic misuse, resulting from a complex interplay actors involving policymakers, healthcare professionals and the general public, underscores the gravity of this issue [9]. These factors are intricately linked to variations in governance, drug regulations, pharmaceutical market structures and cultural influences, all of which contribute to the misuse of antibiotics [10, 11]. Furthermore, a lack of knowledge among healthcare professionals, defensive medical practices, suboptimal patient communication and time constraints during consultations significantly contribute to the inappropriate dispensing of antibiotics [12–15]. User behavior, marked by both misuse and overuse, is heavily influenced by social norms and deeply ingrained beliefs [16–20].

Effective communication is widely recognized as pivotal in healthcare to ensure accurate understanding and successful treatment [21]. The global focus on antibiotic dispensing, which hinges on communication between dispensers and users, has intensified due to its connection with the development of AMR [22–24]. This connection is rooted in the inadequate emphasis on completing the full antibiotic course [25, 26], often caused by insufficient consultation time and ineffective communication skills [27–29], as well as an imbalance in the power dynamic

between healthcare providers and clients, leading to premature cessation of antibiotic use [27, 30].

User beliefs about antibiotics often conflict with the instructions provided, significantly influencing adherence to treatment plans. These beliefs often stem from misunderstandings, fear of side effects and a sense of improvement in health [31]. Misunderstandings manifest when users overestimate their knowledge or lack thereof, leading to antibiotic misuse, such as the belief that antibiotics can effectively treat non-bacterial ailments such as coughs, colds, body pain and flu [32–36]. Fear of side effects and premature discontinuation of antibiotics due to a sense of improvement, despite instructions to the contrary, also influence user behavior [37, 38]. Thus, in this study, beliefs are characterized as misconception, indicating an inaccurate or flawed understanding [39] of how antibiotics function, what side effects are or the reasons behind occurrences.

A systematic review of communication interventions aimed at improving antibiotic use among the public highlighted a dearth of research conducted in Thailand [40]. Among the existing studies, a critical discovery was that improved communication regarding antibiotic knowledge between public health officers and grocery shop owners could reduce the inappropriate sale and use of antibiotics in the community [41]. Another study in Northern Thailand underscored the role of educational activities and local language in enhancing antibiotic attitudes and practices [42].

While studies on communication between health professionals and patients about medications have been a focus for decades [43, 44], limited research in Thailand has encompassed both formal and informal dispensers, as well as users, within the same study. Therefore, this study seeks to comprehensively examine the broader spectrum of dispensers and users, which includes both formal and informal dispensers. It aims to shed light on the comprehensive examination of antibiotic practices and communication within community settings in developing countries such as Thailand.

## Methods

### Study site

This paper reports on results from the study conducted at the Kanchanaburi Health and Demographic Surveillance System (HDSS) in Thailand, which is one of the INDEPTH (International Network for the Demographic Evaluation of Populations and Their Health in Developing Countries) community-based study sites. It is a sub-study of the project on "Community-level antibiotic access and use in low- and middle-income countries: Finding targets for social interventions to improve rational antimicrobial use", known as the ABACUS (AntiBiotic ACcess and USe) study. Its focus is on appropriate access and use of antibiotics between six different health and demographic surveillance sites in low- and middle-income countries in Asia (Bangladesh, Vietnam and Thailand) and Africa (Mozambique, Ghana and South Africa) [45]. For further details, see https://abacus-project.org/project/. The overall aim of the project is to assess and compare community-based antibiotic access and consumption, as well as the underlying factors, in six countries across Africa and Asia. Additionally, the project aims to systematically investigate understandings of antibiotic use, along with related health-seeking behaviors and the broader contextual factors influencing these behaviors. As this sub-study is conducted in Thailand, which is an upper- middle-income country in Asia, it contributes to a better understanding of antibiotic usage.

The Kanchanaburi HDSS is located in western Thailand and was established in the year 2000, with a cohort size of approximately 43,000 in 2004. This study, however, included about 8,000 persons residing in urban and semi-urban areas of the Kanchanaburi HDSS [46]. Two

phases of data collection took place in urban and semi-urban communities of the Kanchanaburi HDSS: 21 July 2016–19 June 2017 for the first phase and 26 August–15 September 2018 for the second phase.

## Data collection techniques

This paper presents the study's qualitative findings, which involved two phases of data collection. During the first phase, 34 in-depth interviews (IDIs) were conducted, involving 18 drug suppliers and 16 community members who use antibiotics. Additionally, six focus group discussions (FGDs) were held with community members. In the second phase, we conducted follow-up IDIs with eight drug suppliers and two more FGDs with community members, addressing important issues identified in the initial phase. The interview and discussion guides are available in the S1 File. Data collection for the IDIs and FGDs was carried out by three researchers (M.S., S.P. and W.J.) who have extensive training and experience in qualitative research and have been actively involved in the Kanchanaburi HDSS since 2000. The core questions used in this study were standardized across all the other INDEPTH ABACUS study sites.

## Sample selection

**In-depth interviews (IDIs).** The key informants for IDI were heads or main responsible persons for drug dispensing and community members located in the Kanchanaburi HDSS. In this study, the key informants for antibiotic dispensers (suppliers) were those who worked at both public and private health facilities. Thailand's healthcare system comprises three levels: primary (sub-district health promoting hospitals), secondary (district, provincial and regional hospitals) and tertiary (university and large private hospitals). Public and private sectors play vital roles, with public facilities serving about three-quarters of the population. Informal healthcare providers such as pharmacies and traditional healers also contribute to the healthcare landscape [47–49] (see the S2 File for further details).

**Drug suppliers.** At the tertiary level, there was one public provincial hospital located in our study area that was willing to participate in this study for the two phases of data collection. Two health facilities were recruited to our study for the secondary level of healthcare service for the first and second phases. There were three private hospitals located in our study area, two of which agreed to participate in the first phase and one in the second phase.

The primary level includes healthcare services in the public and private sectors. The public sector included community health promotion hospitals (CHPs) and local health centers, whereas the private sector included private clinics and drugstores. There were 12 CHPs and local health centers in our study area. According to our qualitative research methodology, designed in addition to budget and time constraints, three key informants from the CHPs were interviewed in depth in the first phase. The criterion to select CHPs was based on the coverage population under each CHP, with the first, second and third ranking being selected. Two CHPs participated in the second data collection phase.

The researchers approached private clinics through local knowledge and our networks by providing relevant research information to the owner. The Kanchanaburi provincial health office reported that there were 101 private clinics operated by different levels of health profession in our study area. The selection criterion for private clinics was based on the willingness of the private clinic owner. The research team was able to recruit two private clinics in the first phase. However, because the second phase involved following up on the issues identified in the first phase with the same drug suppliers, both private clinics declined to participate in this second phase.

The health provincial office provided a full list of licensed pharmacy shops in our study, which comprised 96 shops. A health provincial officer–a pharmacist–contacted drugstore owners because of participants' trust in local authorities. Finally, six drugstore owners agreed to participate in the first data collection phase. One drugstore was selected for IDI for the second phase because it had the highest number of antibiotic encounters. The owners or pharmacists were the IDI key informants.

Although only medical doctors or licensed pharmacists can dispense antibiotics, some local grocery stores sell drugs and antibiotics by laypersons. Based on community leaders' and village health volunteers' networks, two grocery shops were willing to be interviewed in depth in the first phase but not in the second.

In this study, the researchers did not substitute the declined health facilities (private hospital, CHP, private clinic and drugstore) because we wanted to follow up with the interesting issues that came up during the first phase.

**Community members.** Another group of key informants for IDI was the community members, who are antibiotic users. Sample selection was based on a simple random sampling technique. Selection of community members was made randomly from the HDSS database, which aimed to use standardized methods for studying antibiotic use in six low- and middle-income countries. This reflected the full geographical range of the site. A sampling frame was employed from two sources of data; the first source was a list of household members in the Kanchanaburi HDSS database and the second source was a list of clients that received antibiotics from a health-promoting hospital during the year before the IDI. The population in the sampling frame was stratified into five groups: age 18–59 years (male and female), age 60 years or older (male and female) and mothers of children aged 5 years or younger. The researchers employed the SPSS simple random command to select key informants randomly. The 16 community key informants consisted of two males and two females aged 18–59 years, two males and two females aged 60 years or older and eight mothers or guardians of children under 5 years old. The IDI data collection from community members was done only in the first phase.

The information gathered included experience of obtaining and using medicines by community members, focusing on where medicines are obtained, how they are used and for what reasons. In addition, the researchers collected information related to issues of accessing treatment, supplier/sellers of medicines and the medicines per se.

**Focus groups discussions (FGDs).** Eight FGDs were conducted and divided into two phases, six in the first phase and two in the second. Each FGD consisted of six to eight participants. Participants were recruited through the community network, which involved collaboration between the research team and community members in the study areas. The selection of participants was based on the population of the Kanchanaburi HDSS from which the individuals were chosen.

In the first phase, the first four FGDs included two groups of males and two groups of females, with participants aged 18–29 years (two FGDs) and 30 years or older (two FGDs). The other two FGDs included village health volunteers and the elderly.

In the second phase of the study, we focused on common themes that emerged from the first phase. It was determined that two specific groups of participants played a crucial role in shaping these themes: males aged 60 years or older and females aged 18–59 years.

The standardized discussion guideline for the FGD had been used in all sites of the INDEPTH ABACUS study (see the S1 File). These core questions aimed to gather information on accessing treatment, medicines, suppliers/sellers and antibiotic resistance.

## Study procedures

Convenient places and times for the participants were the most important issues for the researchers to conduct IDI and FGD. IDIs were conducted at participants' houses, clinics and workplaces, whereas FGDs were conducted at community centers such as community halls, meeting rooms, pavilions and temples located in the study areas. M.S., S.P. and W.J. conducted all the IDIs, whereas M.S. moderated all the FGDs with assistance from S.P. and W.J. They used the Thai dialect to run the IDI and FGD, with discussion guidelines encouraging dialogue with optional probes. The IDI lasted 45–60 minutes, whereas the FGD took 60–90 minutes. An audio record was used for each IDI and FGD if participants consented. One participant in our study did not permit audio recording of the IDI but alternatively gave consent for note-taking.

## Data analysis

The audio transcription was conducted in Thai to facilitate data analysis, with the analysis itself also carried out in Thai. Line-by-line coding was performed using the OpenCode program [50]. Subsequently, thematic analysis and codes were assigned to the material. M.S. and S.P. determined the thematic saturation. This study applied the 32-item Consolidated Criteria for Reporting Qualitative Studies (COREQ) checklist [51].

## Sample characteristics

There were a total of 88 individuals who participated in the FGD and IDI, including community members and suppliers. Community members included 30 males and 40 females. Most had completed primary school and were aged between 18 and 82 years. A majority of them, almost two-thirds or 60%, were involved in sales and service, working for wages and living in non-poor households. All have access to health services under universal healthcare. Eighteen suppliers participated in this study, including ten females and eight males. Most had completed a bachelor's degree in pharmacy and their ages ranged from 29 to 72 years.

## Results

The results are presented according to the main themes that emerged during the data collection: communication quality; communication and imbalanced power; and misconceptions and instruction. Communication quality emphasizes the dispensing behavior and communication between antibiotic dispensers and users. Communication and imbalanced power focuses on the space linked with the power to control antibiotics. In this study, power is defined as the ability to influence actions, encompassing both 'power over' and 'power to'. 'Power over' denotes control over decision-making, resources, ideas and meanings, whereas 'power to' relates to an individual's ability to take action [52]. Thus, power is observed through the interaction between dispensers and users in health facility and home settings, involving access to and knowledge of antibiotics as well as the utilization of antibiotics. Finally, the theme of misconceptions and instruction reveals the misconceptions that contradict antibiotic instructions, affecting antibiotic adherence.

### Theme 1: Communication quality

This theme focuses on communication between dispensers (suppliers) and users (community members) and is mainly related to communication quality. Both dispensers and users have raised concerns of communication constraints, language use and incomplete information.

Constraints to communication (i.e., comprehension) at different healthcare system levels seem to have similarities and differences. The major constraints to effective communication appeared

at the tertiary, secondary and (less so) primary care levels in the public sector. By contrast, the least constraints appeared at the private health facilities. One explanation may be that the more highly managed and bureaucratic procedures in the public healthcare setting are more complex and confusing for the client than the more service-oriented system in the private setting.

In the public tertiary care setting, the interaction between dispensers and users is constrained by time limitations. This is primarily caused by overcrowding, especially when doctors initiate physical exams simultaneously and prescriptions are concurrently sent to the pharmacy section. As a result, the pharmacy section becomes overwhelmed by a high volume of medical orders, leading to a scarcity of time for providing instructions.

> *". . . in the late morning, when the patients accumulate, there are long queues, and we have to talk fast. Some patients can understand the information, but others may not."*

(IDI, supplier, female, age 40+, public hospital)

The communication constraints observed at the tertiary care level are also prevalent at the public secondary care level. Additionally, one participant noted that communication challenges are not solely attributed to the healthcare system but are influenced by factors such as the patient's time and surrounding environment.

> *"There are many factors (affecting the difficulty in communication) such as when the patients were in a hurry or there are too many patients . . . moreover, it is noisy from the megaphone (calling the "client's name")."*

(IDI, supplier, female, age 40+, public hospital)

At the public primary care level, the atmosphere is generally more laid back, allowing staff to dedicate ample time to interacting with patients. Nonetheless, a challenge arises when communicating with specific groups, such as cross-border migrants who are not of Thai origin. This difficulty arises from their limited understanding of the Thai dialect.

> *"Now we have problems in communicating with migrants from Myanmar. Some can speak Thai a little bit, but some cannot speak at all. Often, there is a supervisor from the workplace of the migrants to serve as translator and advocate. However, we are not sure whether the clients will follow our instructions on proper use of the drugs."*

(IDI, supplier, female, age 40+, public hospital)

In contrast, the private healthcare setting offered greater flexibility in terms of time for patient communication, providing ample opportunity for interaction between healthcare providers and the individuals seeking care. The dispenser employs various strategies at this facility to ensure patient attentiveness toward the provided instructions.

> *"We normally give direct information to the patients. Apart from that, we observe how they respond to our explanation to assess comprehension. When we complete the instructions, we will ask whether they understand how to take the drugs. We also use some techniques to focus their attention, such as asking what types of medicines they are allergic to, or addressing them by their name."*

(IDI, supplier, male, age 50+, private hospital)

The community members discussed that they were given information about antibiotics related to their illness but did not fully understand the explanation. Some felt that the information was insufficient and wanted to know more about antibiotics.

*"When talking about germ killer medicine, we only take it because they told us to, without any explanation. They only tell us that this is a germ killer medicine, but do not explain what types of germs it kills."*

(FGD, community members, male, age 60+, urban)

Apart from time constraints that limit the information received, language use is another issue raised by both the suppliers and community members. The technical terms used by dispensers are often unfamiliar to users: for instance, "antibiotic resistance". This term sometimes leads to confusion among community members. Some of the community members mentioned that sometimes they were given an explanation as to why they had to finish all the antibiotics but they could not understand the term "antibiotic resistance".

*"I do not know. I heard them say "drug resistance". I do not know what they are referring to."*

(IDI, community members, female, age 47, semi-urban)

The community members also needed simple words to understand the content rather than the technical term that they rarely understood. The community members from the FGD mentioned that the technical term confused them, making it hard for them to follow the content.

*"The providers have to use simple words that the villagers are familiar with; then they will understand. For example, I used to invite a nurse to speak during a meeting, and she communicated using simple words, avoiding technical terms to ensure the villagers could comprehend. This made it easier for them to understand the content."*

(FDG, community members, female, age 15–59, semi-urban)

Although time constraints were different between the levels of the healthcare service system, the interpretation of information on antibiotics seemed to be similar. This reflects that the dispensers are familiar with such information. Imbalance of information about antibiotics between dispensers and users occurred, resulting in most users receiving or retaining imperfect information about the use of antibiotics.

*"We need to ask the pharmacist how to take the tablets. If there is more than one pill, do I take them together or at different times? I get confused."*

(FGD, community members, female, age 15–59, semi-urban)

While the importance of providing instructions for antibiotic use was acknowledged, our participants noted that when antibiotics are obtained from grocery stores, there are no accompanying instructions. In this scenario, users simply inform the seller of the specific antibiotic they wish to purchase.

*"They (sellers at grocery shops) do not tell anything about the medicine we bought."*

(FGD, community members, female, age 18–30, semi-urban)

## Theme 2: Communication and imbalanced power

This theme focuses on the relationship between dispensers and users regarding the access and utilization of antibiotics. The problem of accessing antibiotics is closely tied to the choice between public and private healthcare facilities, while the location of administration influences the issue of their usage, whether it occurs in the health facility or home setting. These concerns have been brought up for consideration.

Although they have a health scheme that provides them with affordable or free healthcare services, numerous Thai still pay out of pocket when the prescribed medication was not what they expected. As a result, they opt for private sector services, thereby incurring the entire expense. Some suppliers acknowledge that community members are entitled to free access to medication, yet some individuals even opt for informal alternatives when they are unable to obtain the desired medication.

*"We have concerns about the clients' health scheme, some of them can get service with no cost. Anyway some choose to buy medicines from drugstores. If they do not feel better, they come to see us. If they ask us for a specific medicine which we refuse to dispense, they say that they will just go buy it from the drugstore."*

(IDI, supplier, female, age 40+, public hospital)

Community members have additionally conveyed that in instances where they are unable to obtain antibiotics from public health facilities, they resort to purchasing them from local drugstores.

*"Once I went to see the doctor at the hospital and I did not get antibiotics, I decided to buy Amoxicillin from the drug store because I know that I had tonsillitis."*

(FGD, community members, elderly, age 60+, semi-urban)

Access to antibiotics reflects how power is exercised. At public healthcare facilities, the users have less power to negotiate with dispensers. In other words, they let the healthcare providers be a legitimate interpreter in order to dispense antibiotics to them.

*"The doctor told me that this medicine is a "germ killer". I can understand easily that taking the germ killer will relieve my sickness quickly . . . I never doubted the doctor; I just followed his instruction on how to take the medicine after or before meals. I did not have knowledge on this, so I did whatever they told me to do. However, if I take the medicine and I am not cured, I will buy new medicines from another source."*

(IDI, community members, male, age 45, semi-urban)

By contrast, the clients have some power to bargain when they are at private health outlets. Private health facilities are more patient-oriented, which allows the users to exercise power. For instance, they can ask for medicine if they feel they have another choice.

*"Some clients, they are not happy with amoxicillin, they asked for Augmentin. We need some times to explain to them why the doctor starts with amoxicillin. Some clients are still not satisfied with our explanation, in this case, we have to consult the doctor."*

(IDI, supplier, male, age 50+, private hospital, urban)

Overall, however, it seems that users have the most power at the unlicensed shops that sell antibiotics. Here, they generally get what they demand. Our participants mentioned that sometimes community members just named the medicine they wanted, and these included antibiotics.

*"At the drug store, they will buy these medicines [T-C mycin, penicillin (500,000), ampicillin] for wound and sore throat."*

(FGD, community members, female, age 18–59, semi-urban)

Although the users have less power in the health facility setting, some users gain control of antibiotic use when they take matters into their own hands. After getting antibiotics, some community members reported that the use of antibiotics depends on their own decision once the antibiotics are with them.

*"We got instructions, but sometimes we did not follow them. It depends on ourselves because we are already feeling better."*

(IDI, community members, male, age 60+, semi-urban area)

### Theme 3: Misconceptions and instruction

The main focus is on the contradiction between people's misconceptions and the recommended antibiotic guidelines, leading to a lack of adherence. Although antibiotic practices are influenced by several factors, individuals' misconceptions are one such factor that affects whether they choose to follow the instructions provided or not, which directly affects antibiotic adherence.

Dispensers at formal healthcare facilities, whether public or private, always provide both oral and written instructions to patients or users and mostly advise them to complete the course of antibiotics.

*"For antibiotics, some customers specify their names. I advise them to finish the entire pack; they must take the full course. Some people ask how many days they have to take it. I dispense one pack at a time and advise them to finish it all. If the symptoms are still not completely resolved, then I ask them to come back and receive more medication. Typically, I dispense at least 3 days' worth of amoxicillin and 5 days' worth of antibiotics for patients with urinary tract infections.*

*On the medication bag, I provide the written instruction (i.e., medication for infection and advise taking all of them)."*

(IDI, supplier, female, age 42, pharmacy shop, semi-urban)

Community members have verified that the information provided is consistent in both oral and written forms.

*"Actually, the pharmacist told me how to take the medicine and also mentioned in the written instruction how many times a day and before or after meal."*

(FGD, community members, elderly, age 60+, semi-urban)

However, users stopped taking antibiotics once they experienced relief. In the FGD with village health volunteers, it was a surprise that even among these trained volunteers, who know

that they have to complete the full dose of antibiotics in whatever conditions, only two in seven participants said that they took the complete dose of the antibiotics when they were ill. The others indicated that they stop the antibiotics when they feel better. This shows how ingrained this practice is in the community.

> *Participant 1: I stop, when I feel better*
>
> *Participant 2: I took it for only 2 days*
>
> *Participant 3: When I feel a little bit better, I stop*
>
> *Participant 4: Somebody said the antibiotic is dangerous, I didn't dare to take a lot of it*
>
> (FGD, community members, female, village health volunteers, urban)

In addition to the village health volunteers' views, community members also confirmed their tendency to prematurely discontinue antibiotic usage when they feel improvement. Information gathered from seven FGDs and nine IDIs with community members confirmed that many stopped taking the antibiotics as instructed.

> *"I think only 20 percent of those community members finish all antibiotics as indicated in the instruction."*
>
> (FGD, community members, female, age 18–59, semi-urban)

After getting the antibiotic instructions, whether community members follow the instructions or not is sometimes linked to their level of knowledge. One of the dispensers indicated that knowledge about antibiotics would help in adherence.

> *"They have the knowledge, but each individual has a different level of knowledge. Those who care about their health will seek accurate information and take the antibiotic properly."*
>
> (IDI, supplier, female, age 42, urban)

Although most community members believe in the curative powers of antibiotics for their illnesses, many people try to avoid taking them. This is due to the fact that they fear the side effects and interaction with other medications they take. Thus, they try to take less or stop taking antibiotics as soon as possible and limit overall antibiotic usage.

> *"Q: From the instruction to finish all of them (antibiotics), some of you told me you stopped when you felt better. Do any of you do the same?*
>
> *Participant 6: Yes, when I got rid of sick, I stopped. I am afraid that it may affect the drug I take daily. "*
>
> (FGD, community members, female, age 30+, urban)

The dispenser also corroborated these fears, stating that some community members are apprehensive about the antibiotics' side effects and specifically request fewer such medicines. For instance, one individual expressed concern that antibiotics might harm their kidneys and liver, leading them to request only two antibiotic pills from the pharmacy.

> *"He(customer) told me that he was afraid of it (antibiotics) because it is going to destroy his kidneys and liver. He was really afraid and asked me for only two antibiotic pills."*

(IDI, supplier, female, age 33, pharmacy shop, urban)

## Discussion

This study attempts to understand the dispensers' and users' communication, allowing the researcher to find interaction between the two key factors. Understanding antibiotic dispensing and usage with regard to communication is important for designing and implementing appropriate interventions to address the issue in the broader public health interest.

Our findings point out that constraints in communication occur due to overcrowding and rushed processing of a large number of patients, limiting the time for client education. This situation is particularly prevalent in public tertiary and secondary care settings. Additionally, a lack of communication skills is a significant deficit among some dispensers. These factors may lead to imperfect information exchange and confusion about the indications for antibiotic use. Previous studies found that a shortage of consulting time and poor communication skills led to inadequate explanations for patients and inappropriate antibiotic use [27–29]. The language used in communicating with clients is another concern of our study because it was found that users were less likely to understand the dispenser's language in certain situations. This can lead to misunderstanding about antibiotic use. A previous study suggested that plain and simple language is necessary for effective communication about antibiotics [53]. Furthermore, we should consider specific groups, such as migrants with limited proficiency in the Thai dialect. To address this issue, we propose the use of QR codes containing online information presented in pictograms or the languages spoken by migrants. This approach can effectively convey medication information and enhance comprehension [54], as a previous study found that 87% of migrant workers in Thailand owned smartphones [55].

One of the important findings in our study is that the term "antibiotic resistance" is almost universally misunderstood or unknown among people at the village level. This finding confirms the results of a previous study in Thailand, which found that community residents were not familiar with the term "antibiotic resistance". This may be due to the fact that they never heard, never believed and/or never feared drug resistance [56]. Thus, our findings suggest that it is important to address this issue by providing training or conducting workshops for dispensers to enhance communication skills, particularly communication using lay language. Additionally, integrating QR codes with easily understandable pictogram-based online information, a strategy proven effective in previous research [54], may help to increase antibiotic compliance and overall satisfaction of the users. This approach is appropriate, as the national survey revealed that 84.5% of Thais aged 6 years and over own a smartphone and approximately 82.3% of the population in the study area own a smartphone [57].

Another concern of our study is the practice of buying antibiotics for self-medication for mild illness from grocery stores without instructions. Medications from these shops are mostly dispensed by non-qualified dispensers. This indicates that community members in the study areas can easily obtain antibiotics from individuals who are not qualified dispensers. A previous study found that non-qualified dispensers significantly aggravate the inappropriate use of antibiotics in the community setting [58]. A previous study also found that patients who received antibiotics without a prescription were more likely to stop when their symptoms improved compared to those with a prescription [59]. However, Thailand has a national drug law that allows antibiotics to be dispensed by doctors, pharmacists and health authorities. Ironically, the fact that antibiotics can be bought over the counter without a prescription means that non-qualified sellers can appear qualified to the uninformed user. Thus, antibiotics are

distributed through several channels [60], including legal and illegal distributors [61]. Therefore, a national strategy aiming to improve antibiotic dispensing and reduce non-essential and incomplete use of the drugs also needs to focus on private health facilities, particularly those with unqualified dispensers. This could be one of the effective measures for controlling bacterial resistance. In addition, this finding reinforces that effective instructions, law enforcement and prescription are critical for managing antibiotics in the community setting.

A previous study pointed out that promoting responsible use of antibiotics necessitates acknowledging the significance of power dynamics and autonomy within the healthcare system [26]. Thus, imbalanced power in communication is another concern of our study. The study addresses the issue of imbalanced power in communication, defining power as the ability to influence actions, encompassing both "power over" and "power to". The research observes power dynamics between dispensers and users in both health facility and home settings. This involves access to and knowledge of antibiotics, as well as their utilization. The literature reveals that access to power and resources is negotiated in everyday life [62]. Thus, the health facility setting is not only a clinical space but also a social space in which unequal power relations occur [63]. There is a power imbalance between healthcare providers and clients, particularly in healthcare settings where patients are most likely to be subordinated [27, 30]. Our study found different levels of imbalance of power between antibiotic dispensers and users. At the tertiary and secondary care levels, such an imbalance seems stronger than at the primary and private healthcare levels, where the power of health service providers over users was maintained. This finding reinforces the results of a qualitative study in Australia, where healthcare providers wield power over customers [64]. Similarly, a qualitative study in Spain indicated that service providers have power over users in antibiotic prescribing [65]. The imbalance of power is less in the private sector due to the fact that they are more customer-oriented and less bureaucratic. Notably, the power advantage shifts to the user interacting with the unlicensed dispensers and it can be said that the customer has power over unlicensed dispensers. This aligns with observations in Ghana, where sellers sell antibiotics even when they do not agree with the customers' conditions [66]. In highly regulated and managed healthcare settings such as tertiary and secondary care outlets, users may feel a socio-cultural pressure to behave as "good patients" in the eyes of dispensers, which allows dispensers to have power over users. In that space, the user's role is passive and compliant, and the client is uncomfortable to speak up [27, 67, 68]. Conversely, when the balance of power is more level, the user will feel more confident in speaking up. Not only access to but also knowledge of antibiotics creates a power imbalance. Our study found that users do not necessarily understand the exact nature of their condition and may not understand their treatment options. Thus, they defer their decisions to the dispensers as the legitimated interpreters for their illness.

In contrast, in the home setting of control over antibiotic use, community members have full power to regulate its usage. Exercising this power to use with only limited knowledge of antibiotics may lead to the inappropriate use of these powerful medicines. In addition, lacking knowledge of antibiotics may be a symptom of a more significant health illiteracy problem in parts of society. Health literacy recognizes the ability to understand and critically evaluate health information and make health-related decisions [69]. Previous studies in other parts of the world have also pointed out that education and health literacy play a crucial role in the reduction of inappropriate use of antibiotics [20, 70–74]. Thus, our findings call for enhancing health literacy, which may influence the dispenser–user relationship, self-care and the use of antibiotics. Additionally, it is necessary to train healthcare providers on sharing power, promoting horizontal communication through active listening and building relationships with service users [75–77].

Our findings reveal that the most important instruction for antibiotic use includes how to take the drugs and the need to finish them all. But users are not always told the reason why they have to finish the drugs, which is particularly relevant if they feel cured after a day or two. This lack of complete information and communication discourages compliance and is a primary reason for lack of adherence. The contradiction between misconception and antibiotic instruction, while giving instructions for antibiotic use, operates through the prevailing misconceptions of the community members. This interaction can result in stopping or continuing antibiotic use due to misconception about antibiotics and the fear of side effects, which in turn reflects treatment compliance.

This study found that participants universally received instructions for antibiotic use from formal dispensers, whereas those who got their supply from unlicensed sellers did not receive instructions. The main information from the formal sector included how to take the drugs and the need to finish the complete dose, regardless of how well they felt. Notably, users are not always told the reason why they have to finish the drugs, which is particularly relevant if they feel cured after a day or two. Such lack of information discourages compliance and is a primary reason for lack of adherence.

Moreover, it seems that some community members may not fully embrace these explicit guidelines on antibiotic usage due to concerns about potential side effects. This fear of side effects can lead individuals to prematurely discontinue or modify their antibiotic treatment, even if it contradicts the recommended instructions. A previous study found that misunderstanding of the consequences of unnecessary antibiotic use brought about non-adherence [78]. Nevertheless, concerns about side effects are legitimate and healthcare personnel should listen and respond with empathy and professionalism. This empathetic communication is crucial [79].

In addition to misconceptions, fear also leads individuals to prematurely stop or modify antibiotic treatment, despite contradicting the recommended instructions. A study in Saudi Arabia demonstrated that participants would take less of their antibiotics to avoid adverse effects without consulting a healthcare professional [37]. Our findings shed light on the significance of addressing misconceptions and misunderstandings surrounding antibiotic use to improve adherence rates in the community. This highlights the need for comprehensive education and communication efforts to ensure that individuals understand the rationale behind the antibiotic treatment guidelines and feel more confident in adhering to them, especially regarding when and how to take antibiotics. For instance, taking antibiotics as prescribed and knowing when not to use them, particularly with regard to viral infections [80].

## Strengths and limitations

The main strength of this study is the integration of dispensers and users. Healthcare providers, suppliers and users of antibiotics provided practical insights on how and why these drugs are controlled and dispensed, how users are instructed and how clients use the drugs.

This study also has several limitations. Because antibiotic dispensing is regulated by a national drug law, our study was not able to obtain empirical information from illegal antibiotic dispensers on selling and dispensing by unqualified persons. This limitation calls for further research that deals with those illegal dispensers; perhaps indigenous field worker sampling (IFWS) is required because it allows local community members who have the advantage of reaching the target sample to be trained as investigators [81]. Additionally, data collection was conducted during 2016 and 2017, and the context may have changed since then.

## Conclusions

The evolving practices of antibiotic usage depend on both dispensers and users and their communication. Future interventions should aim to enhance communication between dispensers and users, improve user knowledge and provide clear instructions to facilitate user compliance. Exploring alternative communication channels, such as QR codes, is a viable option. This approach can contribute to reducing one of the major causes of AMR.

## Supporting information

**S1 File. Research tools.**
(PDF)

**S2 File. Healthcare delivery system in Thailand.**
(PDF)

## Acknowledgments

The kind cooperation and sharing of information by all the participants made this study possible. We would like to thank all of them. We would also like to thank Jettapon Sangkla, who assisted us with the data collection. Finally, we would like to express our appreciation to Osman Sankoh, Margaret Gyapong, Johannes John-Langba and John Kinsman, who supported the study.

## Author Contributions

**Conceptualization:** Malee Sunpuwan, Sureeporn Punpuing, Wipaporn Jaruruengpaisan, Heiman Wertheim.

**Data curation:** Malee Sunpuwan, Sureeporn Punpuing, Wipaporn Jaruruengpaisan.

**Formal analysis:** Malee Sunpuwan.

**Funding acquisition:** Sureeporn Punpuing, Heiman Wertheim.

**Investigation:** Malee Sunpuwan, Sureeporn Punpuing, Wipaporn Jaruruengpaisan.

**Methodology:** Malee Sunpuwan, Sureeporn Punpuing, Wipaporn Jaruruengpaisan, Heiman Wertheim.

**Project administration:** Malee Sunpuwan, Sureeporn Punpuing, Wipaporn Jaruruengpaisan.

**Resources:** Malee Sunpuwan, Sureeporn Punpuing, Wipaporn Jaruruengpaisan.

**Software:** Malee Sunpuwan, Wipaporn Jaruruengpaisan.

**Supervision:** Malee Sunpuwan, Sureeporn Punpuing, Heiman Wertheim.

**Validation:** Malee Sunpuwan, Sureeporn Punpuing, Wipaporn Jaruruengpaisan.

**Visualization:** Malee Sunpuwan.

**Writing – original draft:** Malee Sunpuwan, Wipaporn Jaruruengpaisan.

**Writing – review & editing:** Malee Sunpuwan, Sureeporn Punpuing, Wipaporn Jaruruengpaisan, Heiman Wertheim.

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
