## [Decision Letter · Decision Letter 0]

26 Nov 2023

PONE-D-23-33609Understanding antibiotic use in the community setting in Thailand: Does communication matter?PLOS ONE

Dear Dr. Sunpuwan,

Thank you for submitting your manuscript to PLOS ONE. After careful consideration, we feel that it has merit but does not fully meet PLOS ONE’s publication criteria as it currently stands. Therefore, we invite you to submit a revised version of the manuscript that addresses the points raised during the review process.

 Please review the suggestions and comments from both reviewers and address the identified concerns. 

We look forward to receiving your revised manuscript.

Kind regards,

Anselme Shyaka, Ph.D

Academic Editor

PLOS ONE

Journal Requirements:

Did you know that depositing data in a repository is associated with up to a 25% citation advantage (https://doi.org/10.1371/journal.pone.0230416)? If you’ve not already done so, consider depositing your raw data in a repository to ensure your work is read, appreciated and cited by the largest possible audience. You’ll also earn an Accessible Data icon on your published paper if you deposit your data in any participating repository (https://plos.org/open-science/open-data/#accessible-data).

3. Please include a complete copy of PLOS’ questionnaire on inclusivity in global research in your revised manuscript. Our policy for research in this area aims to improve transparency in the reporting of research performed outside of researchers’ own country or community. The policy applies to researchers who have travelled to a different country to conduct research, research with Indigenous populations or their lands, and research on cultural artefacts. The questionnaire can also be requested at the journal’s discretion for any other submissions, even if these conditions are not met.  

Please find more information on the policy and a link to download a blank copy of the questionnaire here: https://journals.plos.org/plosone/s/best-practices-in-research-reporting. 

Please upload a completed version of your questionnaire as Supporting Information when you resubmit your manuscript.

"1. The Wellcome Trust , UK 

 2. The Institute for Population and Social Research, Mahidol University, Thailand"

**Additional Editor Comments:**

Dear Dr. Malee Sunpuwan,

Thank you for submitting your manuscript for publication by PLOS ONE.

Please find the reviewers’ comments and suggestions for your consideration and appropriate action.

I am looking forward to receiving your revised manuscript.

Sincerely,

Reviewers' comments:

Reviewer's Responses to Questions

**Comments to the Author**

1. Is the manuscript technically sound, and do the data support the conclusions?

Reviewer #1: Partly

Reviewer #2: Yes

2. Has the statistical analysis been performed appropriately and rigorously? 

Reviewer #1: N/A

Reviewer #2: N/A

3. Have the authors made all data underlying the findings in their manuscript fully available?

Reviewer #1: Yes

Reviewer #2: Yes

4. Is the manuscript presented in an intelligible fashion and written in standard English?

Reviewer #1: Yes

Reviewer #2: Yes

5. Review Comments to the Author

Reviewer #1: This is a well-written article. The background information and methods used are clearly described, and the results are interpreted in a way that echoes previous research on the same subject. It's a call for better communication in the dispensing of antibiotics.

However, I have minor comments:

p.2, line 30: In the ‘Objective’ section, it should be written ‘to gain a better understanding of the communication around the use’, and not ‘understanding of the use’. You are basing your interpretations on interviews so you cannot really assess antibiotic use, which would have required observational studies.

p.2, line 48: the term ‘beliefs’ seems inappropriate throughout the paper. No definition is provided, and one gets the impression that the authors talk about misconceptions rather than beliefs. It also seems a little patronizing to talk only about the community members’ beliefs– as if healthcare professionals didn’t have beliefs.

p.11, line 266: the term ‘beliefs’ is unclear. Do you mean ‘representations’ ? Because these ‘beliefs’ can come from previous personal experiences, can they only be ‘beliefs’? I strongly recommend that the authors question the use of the term ‘belief’ throughout the document.

p.4, line 90: ‘actors’ should be written instead of ‘factors’, as the authors seem to be referring mainly to individuals or groups, and not to organisations or other types of factors.

p.6, line 134: How is this sub-study linked to the overall study? We would need more information about this.

p.6, line 145: Data collection was carried out in 2016 and 2017, i.e. 6-7 years from now. Are there any reasons for the delay in publishing the results? This fact should be considered as a limitation, as the context may have changed somewhat since then. It would be important for the authors to take this into account in their article.

Methods

p.8, line 185: No private clinics were recruited in the 2nd phase. Why was this?

p.9, line 202: Why was random sampling used to recruit community members? Given the small sample size, other strategies such as snowball sampling with clear selection criteria could have been used. What is the reasoning behind this approach?

p.9, line220: FGD, could you explain a little bit more what you mean by ‘recruited through the community network’?

Study procedures

p.10, line 234: Where did the FGD take place?

Data analysis

p.10, line 242: After the recordings were transcribed, were they translated into English for coding, or was the analysis done in Thai?

Theme 3

p.17, line 415-416: To discuss the link between people’s “beliefs” and their practices regarding the use of antibiotics as being solely causal is rather simplistic and reductive. The influence of a person’s beliefs on their practices is not so simple; other factors may come into play. In the sentence line 416-417 “Antibiotic practices are influenced by individuals’ beliefs”, I should mention that these “beliefs” are not the only factors.

When the participants talk about their fear of side effects and interaction with other medications they are taking, we (as researchers or health professionals) need to take this seriously. These are legitimate concerns we need to listen to, and not some ‘beliefs’ we have to get rid of. If we deny people’s concerns and needs, we will continue to reproduce a very authoritarian and biomedical way of thinking about antibiotic use.

p.21, line 500: what do you mean by ‘at the village level’? It doesn’t seem very clear to me.

p.24, line 527: In this sentence, the example does not refer to a ‘belief’ but rather to the fact that users were not given access to the information they needed, which is quite different from what you write. I recommend changing this sentence.

Reviewer #2: Thanks for an interesting and well-written manuscript, very well done!

I would suggest changing the word “authoritative” in the abstract (line 54) as it seems like a rather “top-down” approach to change behaviors in relation to antibiotic use.

Even though you have attached a document (S2 File), explaining the different healthcare service levels (tertiary, secondary and primary), I would suggest that you include a brief description about this (like 1-2 sentences) in the article to improve readability.

It would be helpful if you could provide some additional information in the text about the community; how do people generally live, occupation, class (rich, poor etc.?), general access to services? In relation to your suggestion on line 496 about QR codes, do people in rural areas typically own a smart phone, for example?

You mention “power” several times in the article, but without providing a clear definition or any references to other scholars that engage with concepts of power. As there are numerous definitions of power in the wider social sciences literature, please include a brief definition of power in the article.

I have highlighted a few sentences/words in the paper that I found difficult to follow, where I would suggest some editing for improved readability.

Please add a reference on line 533 as you have previously stated that: “The literature reveals that access to power and resources is negotiated in everyday life”. What specific literature do you refer to here?

6. PLOS authors have the option to publish the peer review history of their article (what does this mean?). If published, this will include your full peer review and any attached files.

Reviewer #1: No

Reviewer #2: No

---

## [Author Response · Author response to Decision Letter 0]

17 Jan 2024

January 17, 2024

Anselme Shyaka 

Academic Editor

PLoS ONE

Dear editor,

We appreciate the opportunity to resubmit our revised manuscript and would like to extend our gratitude to both you and the reviewers for their positive feedback and constructive comments. We have carefully considered and incorporated the suggested corrections and modifications, and we are delighted to resubmit our manuscript for further consideration.

In response to the detailed suggestions provided by you and the reviewers, we have made significant edits aimed at addressing all identified issues and concerns. The point-by-point response to the questions and comments is outlined at the end of this letter to assist in the review of our revisions.

We would like to express our sincere thanks for the invaluable comments and queries that have allowed us to strengthen our manuscript. We have diligently worked to incorporate your feedback, and we hope that these revisions demonstrate our commitment to improving the quality of our submission.

Thank you once again for this opportunity, and we look forward to the possibility of our manuscript being accepted for publication.

Sincerely yours,

Malee Sunpuwan

(Malee Sunpuwan)

Corresponding author

malee.sun@mahidol.ac.th

+662 4410201 ext.604

Journal Requirements:

RESPONSE: Thank you for this suggestion. Our manuscript has been prepared in accordance with the provided guidelines.

Did you know that depositing data in a repository is associated with up to a 25% citation advantage (https://doi.org/10.1371/journal.pone.0230416)? If you’ve not already done so, consider depositing your raw data in a repository to ensure your work is read, appreciated and cited by the largest possible audience. You’ll also earn an Accessible Data icon on your published paper if you deposit your data in any participating repository (https://plos.org/open-science/open-data/#accessible-data).

RESPONSE: Thank you for sharing this valuable information! We were not aware of the significant citation advantage associated with depositing data in a repository. We will definitely consider this for our future work to ensure broader visibility and impact. The idea of earning an Accessible Data icon on published papers is intriguing too.

3. Please include a complete copy of PLOS’ questionnaire on inclusivity in global research in your revised manuscript. Our policy for research in this area aims to improve transparency in the reporting of research performed outside of researchers’ own country or community. The policy applies to researchers who have travelled to a different country to conduct research, research with Indigenous populations or their lands, and research on cultural artefacts. The questionnaire can also be requested at the journal’s discretion for any other submissions, even if these conditions are not met. 

Please find more information on the policy and a link to download a blank copy of the questionnaire here: https://journals.plos.org/plosone/s/best-practices-in-research-reporting. 

Please upload a completed version of your questionnaire as Supporting Information when you resubmit your manuscript.

RESPONSE: Thank you very much for this information. We have completed the questionnaire on inclusivity in global research and attached it as a supporting file.

RESPONSE: Thank you for this suggestion. We have provided the grant numbers as follows: 219403/Z/19/Z.

"1. The Wellcome Trust , UK 

 2. The Institute for Population and Social Research, Mahidol University, Thailand"

Please include this amended Role of Funder statement in your cover letter; we will change the online submission form on your behalf

RESPONSE: Thank you for this suggestion. We have stated that "The funders had no role in the study design, data collection and analysis, decision to publish, or preparation of the manuscript."

RESPONSE: Thank you for the suggestion. We have included captions for Supporting Information files at the end of the manuscript and updated in-text citations, as well as references, to match accordingly.

RESPONSE: Thank you for your suggestion. We have confirmed that none of the papers in our references have been retracted. To verify this information, we utilized the Zotero program, integrated with the Retraction Watch Database. Additionally, we have included some references to support the definition of power and its related studies

Comments to the Author

1. Is the manuscript technically sound, and do the data support the conclusions?

Reviewer #1: Partly

Reviewer #2: Yes

2. Has the statistical analysis been performed appropriately and rigorously?

Reviewer #1: N/A

Reviewer #2: N/A

3. Have the authors made all data underlying the findings in their manuscript fully available?

Reviewer #1: Yes

Reviewer #2: Yes

4. Is the manuscript presented in an intelligible fashion and written in standard English?

Reviewer #1: Yes

Reviewer #2: Yes

5. Review Comments to the Author

Reviewer #1: This is a well-written article. The background information and methods used are clearly described, and the results are interpreted in a way that echoes previous research on the same subject. It's a call for better communication in the dispensing of antibiotics.

However, I have minor comments:

p.2, line 30: In the ‘Objective’ section, it should be written ‘to gain a better understanding of the communication around the use’, and not ‘understanding of the use’. You are basing your interpretations on interviews so you cannot really assess antibiotic use, which would have required observational studies.

RESPONSE: We appreciate your suggestion, and we agree with it. We have modified the objective to read: 'This study aims to gain a deeper understanding of communication around the use of antibiotics in the community and seeks potential implementation strategies to change dispensers' and users' practices in communication aspects' (see line 29)

p.2, line 48: the term ‘beliefs’ seems inappropriate throughout the paper. No definition is provided, and one gets the impression that the authors talk about misconceptions rather than beliefs. It also seems a little patronizing to talk only about the community members’ beliefs– as if healthcare professionals didn’t have beliefs.

RESPONSE: Thank you for this suggestion; we agree to use 'misconceptions' instead of 'beliefs' throughout the manuscript when appropriate.

p.11, line 266: the term ‘beliefs’ is unclear. Do you mean ‘representations’ ? Because these ‘beliefs’ can come from previous personal experiences, can they only be ‘beliefs’? I strongly recommend that the authors question the use of the term ‘belief’ throughout the document.

RESPONSE:Thank you for this suggestion; we have concurred with it and have opted to use 'misconceptions' instead of 'beliefs

p.4, line 90: ‘actors’ should be written instead of ‘factors’, as the authors seem to be referring mainly to individuals or groups, and not to organisations or other types of factors.

RESPONSE: Thank you for this suggestion; we have agreed and used 'actors' instead of 'factors (see lines 83)

p.6, line 134: How is this sub-study linked to the overall study? We would need more information about this.

RESPONSE: Thank you for this suggestion; we have added more information on how this sub-study is connected to the overall study. "and consumption, as well as the underlying factors, in six countries across Africa and Asia. Additionally, the project aims to systematically investigate understandings of antibiotic use, along with related health-seeking behaviors and the broader contextual factors influencing these behaviors. As this sub-study is conducted in Thailand, an upper-middle-income country in Asia, it contributes to a better understanding of antibiotic usage". (see lines 138-140).

p.6, line 145: Data collection was carried out in 2016 and 2017, i.e. 6-7 years from now. Are there any reasons for the delay in publishing the results? This fact should be considered as a limitation, as the context may have changed somewhat since then. It would be important for the authors to take this into account in their article.

RESPONSE: Thank you for this suggestion; we have taken it into consideration and included more information on limitations (see lines 636-637). The reason for the delay is that we continued our research on ongoing issues until 2022, and communication challenges persisted during the latest phase of the study (ABACUS II). However, it was not the primary focus then, so we decided to utilize data from ABACUS I.

Methods

p.8, line 185: No private clinics were recruited in the 2nd phase. Why was this?

RESPONSE: Thank you for this comment, we appreciate your comment and have taken it into consideration. We have now included additional information explaining the reasons for not recruiting private clinics in the second phase. Specifically, the explanation states, 'However, since the second phase involved following up on the issues identified in the first phase with the same drug suppliers, the two private clinics declined to participate in this phase (see lines 189-191) 

p.9, line 202: Why was random sampling used to recruit community members? Given the small sample size, other strategies such as snowball sampling with clear selection criteria could have been used. What is the reasoning behind this approach?

RESPONSE: Thank you for this question. Since this is the sub-study and one of its objective is to compare across sites in six LMICs we have added the information to make it clear “The selection of community members was made randomly from the HDSS database, which aimed to use standardized methods for studying antibiotic use in six LMICs. This reflected the full geographical range of the site” (see line 207-210).

p.9, line220: FGD, could you explain a little bit more what you mean by ‘recruited through the community network’?

RESPONSE: Thank you; we have added more information to clarify 'recruited through the community network’ by including the following: 'Participants were recruited through the community network, which involved collaboration between the research team and community members in the study areas.” (see lines 226-228). 

Study procedures

p.10, line 234: Where did the FGD take place?

RESPONSE: Thank you for this question; we added information on places of conducting IDIs and FGDs 'IDIs were conducted at participants’ houses, clinics, and workplaces, while FGDs were conducted at community canters such as community halls, meeting rooms, pavilions, and temples located in the study areas” (see lines 241-243).

Data analysis

p.10, line 242: After the recordings were transcribed, were they translated into English for coding, or was the analysis done in Thai?

RESPONSE: Thank you for this question. The audio transcription was done in Thai to facilitate data analysis, and the analysis itself was also carried out in Thai (see lines 251-252).

Theme 3

p.17, line 415-416: To discuss the link between people’s “beliefs” and their practices regarding the use of antibiotics as being solely causal is rather simplistic and reductive. The influence of a person’s beliefs on their practices is not so simple; other factors may come into play. In the sentence line 416-417 “Antibiotic practices are influenced by individuals’ beliefs”, I should mention that these “beliefs” are not the only factors.

RESPONSE: Thank you for this point. We agreed that it is not only one factor but also other factors, and we have changed it to. “Although antibiotic practices are influenced by several factors, individuals' misconceptions are one of such factor that affects whether they choose to follow the provided instructions or not.” (see lines 434-437)

When the participants talk about their fear of side effects and interaction with other medications they are taking, we (as researchers or health professionals) need to take this seriously. These are legitimate concerns we need to listen to, and not some ‘beliefs’ we have to get rid of. If we deny people’s concerns and needs, we will continue to reproduce a very authoritarian and biomedical way of thinking about antibiotic use.

RESPONSE: Thank you for this concern. We have considered this suggestion and added more information in the discussion section: 'Nevertheless, concerns about side effects are legitimate, and healthcare personnel should listen and respond with empathy and professionalism. This empathetic communication is crucial (see lines 614-616)

p.21, line 500: what do you mean by ‘at the village level’? It doesn’t seem very clear to me.

RESPONSE: Thank you for raising this question. To provide clarity, we have revised it to say, 'This shows how ingrained this practice is in the community. (see line 460).

p.24, line 527: In this sentence, the example does not refer to a ‘belief’ but rather to the fact that users were not given access to the information they needed, which is quite different from what you write. I recommend changing this sentence.

RESPONSE: Thank you for this suggestion. We have modified the sentence from 'This belief discourages compliance and is a primary reason for the lack of adherence' to 'This lack of complete information and communication discourages compliance and is a primary reason for the lack of adherence' (see line 596-597)

Reviewer #2: Thanks for an interesting and well-written manuscript, very well done!

I would suggest changing the word “authoritative” in the abstract (line 54) as it seems like a rather “top-down” approach to change behaviors in relation to antibiotic use.

RESPONSE: Thank you for this suggestion. We agreed and have replaced the word 'authoritative' with 'trustworthy”. (line 53)

Even though you have attached a document (S2 File), explaining the different healthcare service levels (tertiary, secondary and primary), I would suggest that you include a brief description about this (like 1-2 sentences) in the article to improve readability.

RESPONSE: Thank you; we have included a brief description: 'Thailand's healthcare system comprises three levels—primary (sub-district health-promoting hospitals), secondary (district, provincial, and regional hospitals), and tertiary (university and large private hospitals). Both public and private sectors play vital roles, with public facilities serving about three-fourths of the population. Informal healthcare providers, such as pharmacies and traditional healers, also contribute to the healthcare landscape (more details in S2 File).' (see lines 164-169)"

It would be helpful if you could provide some additional information in the text about the community; how do people generally live, occupation, class (rich, poor etc.?), general access to services? In relation to your suggestion on line 496 about QR codes, do people in rural areas typically own a smart phone, for example?

RESPONSE: Thank you for this suggestion. We have incorporated more information about community characteristics. For instance, "A majority of them, almost two-thirds or 60%, are involved in sales and service, working for wages, and living in non-poor households. All have access to health services under universal healthcare" (see lines 266-268).

Additionally, we have included more statistics about smartphone ownership among migrants and the overall population in Thailand and the study areas (see lines 517-519, 529-531).

You mention “power” several times in the article, but without providing a clear definition or any references to other scholars that engage with concepts of power. As there are numerous definitions of power in the wider social sciences literature, please include a brief definition of power in the article.

RESPONSE: Thank you; we have defined it. In our study, power is described as the ability to influence actions, encompassing both 'power over' and 'power to.' 'Power over' denotes control over decision-making, resources, ideas, and meanings, while 'power to' relates to an individual's ability to take action” (see lines 276-279). Additionally, we have added more information related to the definition of power and related literature in the discussion part (see lines 553-571)."

I have highlighted a few sentences/words in the paper that I found difficult to follow, where I would suggest some editing for improved readability.

RESPONSE: Thank you for the suggestion. We have sent the manuscript to a professional editing and proofreading service. Please find attached the certificate of editing and proofreading.

Please add a reference on line 533 as you have previously stated that: “The literature reveals that access to power and resources is negotiated in everyday life”. What specific literature do you refer to here? 

RESPONSE: Thank you for your careful review. We have cited the following reference:

Ekström M, Stevanovic M. Conversation analysis and power: examining the descendants and antecedents of social action. Frontiers in Sociology. 2023;8:1196672.

6. PLOS authors have the option to publish the peer review history of their article (what does this mean?). If published, this will include your full peer review and any attached files.

Do you want your identity to be public for this peer review? For information about this choice, including consent withdrawal, please see our Privacy Policy.

Reviewer #1: No

Reviewer #2: No

---

## [Decision Letter · Decision Letter 1]

2 Feb 2024

Understanding antibiotic use in the community setting in Thailand: Does communication matter?

PONE-D-23-33609R1

Dear Dr. Sunpuwan,

We’re pleased to inform you that your manuscript has been judged scientifically suitable for publication and will be formally accepted for publication once it meets all outstanding technical requirements.

Kind regards,

Anselme Shyaka, Ph.D

Academic Editor

PLOS ONE

Additional Editor Comments (optional):

Reviewers' comments:

Reviewer's Responses to Questions

**Comments to the Author**

1. If the authors have adequately addressed your comments raised in a previous round of review and you feel that this manuscript is now acceptable for publication, you may indicate that here to bypass the “Comments to the Author” section, enter your conflict of interest statement in the “Confidential to Editor” section, and submit your "Accept" recommendation.

Reviewer #2: All comments have been addressed

2. Is the manuscript technically sound, and do the data support the conclusions?

Reviewer #2: Yes

3. Has the statistical analysis been performed appropriately and rigorously? 

Reviewer #2: N/A

4. Have the authors made all data underlying the findings in their manuscript fully available?

Reviewer #2: Yes

5. Is the manuscript presented in an intelligible fashion and written in standard English?

Reviewer #2: Yes

6. Review Comments to the Author

Reviewer #2: (No Response)

7. PLOS authors have the option to publish the peer review history of their article (what does this mean?). If published, this will include your full peer review and any attached files.

Reviewer #2: No

---

## [Editor Report · Acceptance letter]

22 Mar 2024

PONE-D-23-33609R1 

PLOS ONE

Dear Dr. Sunpuwan, 

I'm pleased to inform you that your manuscript has been deemed suitable for publication in PLOS ONE. Congratulations! Your manuscript is now being handed over to our production team.

Kind regards, 

on behalf of

Dr. Anselme Shyaka 

Academic Editor

PLOS ONE